# ActiveDPO: Active Direct Preference Optimization for Sample-Efficient Alignment

**Xiaoqiang Lin**[1]    **Arun Verma**[2*]    **Zhongxiang Dai**[3]
**Daniela Rus**[4,2]    **See-Kiong Ng**[5]    **Bryan Kian Hsiang Low**[1,2]

[1]Department of Computer Science, National University of Singapore, Singapore
[2]Singapore-MIT Alliance for Research and Technology Centre, Singapore
[3]The Chinese University of Hong Kong, Shenzhen, China
[4]CSAIL, Massachusetts Institute of Technology, USA
[5]Institute of Data Science, National University of Singapore, Singapore
`xiaoqiang.lin@u.nus.edu, arun.verma@smart.mit.edu,`
`daizhongxiang@cuhk.edu.cn,`
`rus@csail.mit.edu, seekiong@nus.edu.sg, lowkh@comp.nus.edu.sg`

## Abstract

The recent success in using human preferences to align large language models (LLMs) has significantly improved their performance in various downstream tasks, such as question answering, mathematical reasoning, and code generation. However, achieving effective LLM alignment depends on high-quality datasets of human preferences. Collecting these datasets requires human preference annotation, which is costly and resource-intensive, necessitating efficient active data selection methods. Existing methods either lack a strong theoretical foundation or depend on restrictive assumptions about the reward function, such as linear latent reward functions. To this end, we propose an algorithm, ActiveDPO, that uses a theoretically grounded data selection criterion for non-linear reward functions while directly leveraging the LLM itself to parameterize the reward model used for active data selection. As a result, ActiveDPO explicitly accounts for the LLM's influence on data selection, unlike methods that select data without considering the LLM that is being aligned, thereby leading to more effective and efficient data collection. Our extensive experiments demonstrate that ActiveDPO outperforms existing methods across various models and real-world preference datasets.

## 1 Introduction

Large language models (LLMs) (Google, 2023; OpenAI, 2023; Touvron et al., 2023; Anthropic, 2023) have demonstrated impressive performance across various tasks, including question-answering (Taori et al., 2023), mathematical reasoning (Wei et al., 2022), code generation (Chen et al., 2021), and many others (Zhao et al., 2023). However, LLMs often fall short when required to produce responses that conform to specific formats or align with human values (Ji et al., 2023; Anwar et al., 2024). To address this, methods such as Reinforcement Learning from Human Feedback (RLHF) (Ouyang et al., 2022; Bai et al., 2022) and Direct Preference Optimization (DPO) (Rafailov et al., 2023) use binary preference feedback collected from human annotators, who indicate which of two LLM responses they prefer, to better align LLM outputs with human preferences in real-world applications. Both RLHF and DPO require high-quality human preference datasets to achieve effective LLM alignment. However, collecting these datasets requires skilled human annotators, making this process both costly and resource-intensive (Liu et al., 2024; Carvalho Melo et al., 2024; Muldrew et al., 2024).

To overcome these challenges, recent works (Mehta et al., 2023; Das et al., 2024; Liu et al., 2024; Muldrew et al., 2024) have proposed methods for actively selecting a smaller subset of preference data (i.e., triplets consisting of a prompt and two responses) for human preference annotation while maintaining alignment performance. Specifically, several recent works (Liu et al., 2024; Muldrew et al., 2024) have proposed heuristic methods for actively selecting preference data to collect human

---

preference feedback. However, these methods lack a rigorous theoretical foundation and therefore do not guarantee reliable performance across different tasks and LLMs (see Fig. 1 in Section 4). In contrast, some works (Mehta et al., 2023; Das et al., 2024) have developed methods with theoretical guarantees to achieve sample-efficient LLM alignment. However, these methods require strong assumptions about the underlying latent reward function (e.g., linearity), which may not hold in the context of LLM alignment. Furthermore, another potential limitation of some existing works (Mehta et al., 2023; Das et al., 2024; Liu et al., 2024; Muldrew et al., 2024) is their dependence on a separate reward model or a selection method that works independently of the LLM being aligned. As a result, these methods often *cannot directly account for how data selection affects the LLM itself.*

These limitations naturally lead to the following question: ***How can we develop an active preference data selection algorithm that is both theoretically grounded and practically effective?*** To answer this, we propose ACTIVEDPO, a novel active preference data selection algorithm. ACTIVEDPO is built on DPO, which has shown comparable or superior empirical performance to RLHF while avoiding the complexity of reward model training and the reinforcement learning process, making it a compelling choice for aligning LLMs with human preferences (Rafailov et al., 2023). Furthermore, ACTIVEDPO uses a theoretically grounded criterion for selecting preference data for complex, non-linear reward functions, while using the LLM itself as a reward model to guide that selection.

Specifically, we establish an upper bound on the error in estimating the reward difference between any pair of responses and their ground-truth reward for a given prompt, expressed in terms of the *gradient of the neural network, which represents the current estimate of the DPO reward function associated with the aligned LLM* (Proposition 1 in Section 3). This result enables us to *leverage the LLM's gradient* to derive an uncertainty measure as a criterion for preference data selection, thereby *explicitly accounting for the LLM's influence* on the data selection process. To improve the efficiency and practicality of ACTIVEDPO, we introduce novel techniques, such as batch selection and random projection with LoRA gradients (more details are given in Section 3.3) to reduce computational cost and storage requirements. These additional techniques make ACTIVEDPO both theoretically grounded and practically effective. Finally, extensive experiments demonstrate that ACTIVEDPO consistently outperforms existing methods across various LLMs and datasets.

The key contributions can be summarized as follows:

- In Section 3, we propose a novel algorithm, ACTIVEDPO, that uses a theoretically grounded active preference data selection criterion for LLM alignment. By leveraging an implicit reward function parameterized by the LLM itself, ACTIVEDPO ensures that the selected preference data is better suited to the specific LLM being aligned.
- In Section 3.3, we introduce techniques such as batch selection and random gradient projection to reduce the computational and storage requirements of ACTIVEDPO, making it more practical for large-scale models.
- In Section 4, we empirically demonstrate that ACTIVEDPO achieves efficient and effective active preference learning across diverse LLMs and datasets.

## 2 PROBLEM SETTING

In LLM alignment, we start with a preference dataset $D$ in which each data point contains a triplet $(x, y_1, y_2)$ where $x \in \mathcal{X}$ is a prompt and $y_1, y_2 \in \mathcal{Y}$ are two responses (which can be written by humans or generated from LLMs). The $\mathcal{X}$ and $\mathcal{Y}$ are the prompt space and the response space, respectively. Denote $n$ as the number of data points in $D$. We aim to find a $k$-sized subset $D^s \subseteq D$ and ask human annotators to provide binary preference feedback on the responses denoted as $y_w \succ y_l \mid x$ where $y_w$ and $y_l$ denote the preferred and rejected response, respectively. Note that $y$ is not the human-preference label but the corresponding response to the prompt. We train the LLM to generate responses that better align with human preference on the labeled data subset $D^l$ using DPO. The objective is to obtain an LLM that gives the most desirable responses (defined by win-rate and reward score as we will discuss later) within the fixed labeling budget of $k$.

**Direct preference optimization (DPO).** We first start by discussing the DPO method, as introduced in Rafailov et al. (2023). DPO begins with training an LLM via supervised fine-tuning (SFT) on a carefully curated, high-quality dataset tailored to a specific downstream task, resulting in a model denoted by $\pi_{\text{SFT}}$. The objective of the SFT is to enable the LLM to effectively follow instructions for

a specific downstream task. Let $\pi_\theta(y \mid x)$ denote the conditional probability of generating $y$ given the prompt $x$, where the model is parameterized by $\theta$. An implicit reward function is defined as follows:

$$r_\theta(x, y) = \beta \left( \log \frac{\pi_\theta(y \mid x)}{\pi_{\text{ref}}(y \mid x)} + Z(x) \right),$$

where $\pi_{\text{ref}}$ is the reference LLM, usually chosen to be the SFT LLM $\pi_{\text{SFT}}$, $\beta$ is the KL-regularization coefficient, and $Z(\cdot)$ is a partition function. Based on this implicit reward function, DPO uses the Bradley-Terry-Luce (BTL) model to model preference feedback. Specifically, BTL assumes that the probability of response $y_1$ being preferred over $y_2$, conditioned on the prompt $x$, is given by:

$$p(y_1 \succ y_2 \mid x) = \frac{\exp\left(r_\theta(x, y_1)\right)}{\exp\left(r_\theta(x, y_1)\right) + \exp\left(r_\theta(x, y_2)\right)} = \sigma\left(r_\theta(x, y_1) - r_\theta(x, y_2)\right), \tag{1}$$

where $\sigma(x) = 1/(1+\exp(-x))$. DPO uses the following training objective to train the LLM:

$$L_{\text{DPO}}(\pi_\theta, \pi_{\text{ref}}) = -\mathbb{E}_{(x, y_w, y_l) \sim D^l} \left[ \log \sigma\left(r_\theta(y_w \mid x) - r_\theta(y_l \mid x)\right) \right]. \tag{2}$$

# 3 METHODOLOGY

**Overview of ACTIVEDPO.** ACTIVEDPO starts with generating responses from an initial data $D$, which consists of instructions/prompts tailored to a specific task. We use the initial LLM model (i.e., $\pi_{\text{SFT}}$) to obtain the responses to form the dataset $D_t$, which forms the pool of selection (Section 3.1). After that, we select a batch of triplets $(x, y_1, y_2)$ with size $b$ according to our selection criterion (Section 3.2). Then, we ask the human annotator to provide preference feedback on the responses for the selected batch of data to obtain the labeled dataset. Finally, we train the LLM with the DPO training objective on the newly labeled dataset. We run this process for $T$ iterations and obtain a final trained LLM that generates responses aligned with human preferences.

---

**ACTIVEDPO** Active Direct Preference Optimization

---
1: **Input:** Initial dataset $D$; Reference LLM $\pi_{\text{ref}} = \pi_{\text{SFT}}$; Initial LLM $\pi_{\theta_0} = \pi_{\text{SFT}}$; parameterized by $\theta_0$; Iteration $T$; Batch size $B$;
2: **for** $t = 1, \ldots, T$ **do**
3:     Generate $m$ pairs of responses from previous LLM $y_1, y_2 \sim \pi_{\theta_{t-1}}(y \mid x)$ for each $x \in D$ to obtain the dataset $D_t$.
4:     $D_t^s = \emptyset$
5:     **for** $b = 1, \ldots, B$ **do**
6:         Select the $(x_b^t, y_{b,1}^t, y_{b,2}^t)$ using Eq. (3)
7:         $D_t^s = D_t^s \cup \{(x_b^t, y_{b,1}^t, y_{b,2}^t)\}$
8:         Update $V_{t-1}$ according to Eq. (4).
9:     **end for**
10:    Obtain the preference feedback $y_w \succ y_l \mid x$ for each data point in $D_t^s$ to get the labeled dataset $D_t^l$
11:    Update the LLM $\pi_{\theta_{t-1}}$ using $D_t^l$ with the DPO training objective in Eq. (2) to obtain $\pi_{\theta_t}$
12: **end for**
13: Return the trained LLM $\pi_{\theta_T}$

---

## 3.1 GENERATION OF THE PROMPT-RESPONSES DATASET

In each iteration of ACTIVEDPO, we regenerate the responses for each instruction/prompt in the dataset for two main reasons. First, while some tasks already have responses written by humans or generated by powerful LLMs, most do not have good responses for each instruction at the start. Generating responses is necessary for these tasks before asking human annotators to provide preference feedback on these responses. Second, even though some tasks already have responses for each instruction, however, these responses are not updated as the LLM improves over time. This is undesirable, since the LLM will not be able to learn to generate better responses (relative to the responses in the original dataset) as it improves. Consequently, we generate new responses for all instructions using the latest model obtained from ACTIVEDPO, enabling ACTIVEDPO training to further improve the LLM through progressively higher-quality responses.

## 3.2 SELECTION OF DATA TO GET HUMAN PREFERENCE ANNOTATIONS

The selection strategy of our ACTIVEDPO is inspired by principled neural dueling bandits (Verma et al., 2025a), which derives uncertainty quantification of the human preference for the latent non-linear reward function estimated using a neural network (NN). Inspired by this, we quantified uncertainty in human preferences for our LLM trained with DPO and demonstrated its empirical effectiveness in Section 4. Consequently, our selection strategy is theoretically grounded and demonstrates empirical effectiveness, unlike the heuristic-based method (Muldrew et al., 2024).

**Proposition 1** (Estimation error of the reward difference (informal version of Proposition 2)). *Let $r_\theta$ denote a fully connected neural network with a width of $m$ in each layer and depth of $L$. Let $\delta \in (0,1)$. Assume that there is a ground truth reward function $r$ and that human preference is sampled from BTL preference modeling. As long as $m \geq M$, then with a probability of at least $1 - \delta$,*

$$\left| \left[ r_{\theta_{t-1}}(x, y_1) - r_{\theta_{t-1}}(x, y_2) \right] - \left[ r(x, y_1) - r(x, y_2) \right] \right| \leq$$

$$\nu_T \| \frac{1}{m} (\nabla r_{\theta_{t-1}}(x, y_1) - \nabla r_{\theta_{t-1}}(x, y_2)) \|_{V_{t-1}^{-1}} + \varepsilon$$

*for all $x \in \mathcal{X}$ and $y_1, y_2 \in \mathcal{Y}, t \in [T]$ when using the DPO objective defined in Eq. (2) with an additional regularization term to train this reward function $r_{\theta_{t-1}}$. $V_{t-1} = \sum_{p=1}^{t-1} \sum_{x, y_1, y_2 \sim D_p^s} \varphi_{t-1}(x, y_1, y_2) \varphi_{t-1}(x, y_1, y_2)$ and $\varphi_{t-1}(x, y_1, y_2) = \frac{1}{\sqrt{m}} (\nabla r_{\theta_{t-1}}(x, y_1) - \nabla r_{\theta_{t-1}}(x, y_2))$. The definition of $M, \nu_T, \varepsilon$ can be found in the Appendix A.*

Proposition 1 is based on the theoretical results from neural dueling bandits (Verma et al., 2025a). This result suggests that if $\| \frac{1}{m} (\nabla r_{\theta_{t-1}}(x, y_1) - \nabla r_{\theta_{t-1}}(x, y_2)) \|_{V_{t-1}^{-1}}$ is smaller, the estimation error of the reward difference will be smaller. Note that the reward difference directly decides the human preference according to the BTL preference modeling as shown in Eq. (1). Consequently, the reward function $r_{\theta_{t-1}}$ will have a more accurate estimation of the human preference on the two responses $y_1, y_2$ given $x$. On the other hand, if $\| \frac{1}{m} (\nabla r_{\theta_{t-1}}(x, y_1) - \nabla r_{\theta_{t-1}}(x, y_2)) \|_{V_{t-1}^{-1}}$ is large, this indicates that the reward model will potentially have an inaccurate estimation of the human preference for the responses and hence a higher uncertainty on the human preference. Therefore, a natural selection criterion arises with the uncertainty defined in Proposition 1. Based on this selection criterion, our selection strategy selects a triplet context and pair of arms $(x, y_1, y_2)$ as follows:

$$x, y_1, y_2 = \text{argmax}_{x, y_1, y_2 \sim D_t \setminus D_t^s} \| \nabla r_{\theta_{t-1}}(x, y_1) - \nabla r_{\theta_{t-1}}(x, y_2) \|_{V_{t-1}^{-1}} \tag{3}$$

The selection strategy in Eq. (3) uses the implicit reward function $r_{\theta_{t-1}}$ which is parameterized by the current LLM $\pi_{\theta_{t-1}}$. Note that we remove $1/\sqrt{m}$ from the selection criterion and $\varphi_{t-1}$ since it only affects the scale of the gradient, and the depth $m$ is not well-defined for the LLM. The selection criterion quantifies the uncertainty in the current implicit reward function, specifically the human preferences for the responses $y_1$ and $y_2$. Specifically, a larger value of the selection criterion in Eq. (3) means that the prompt-response triplet $(x, y_1, y_2)$ is more different from the previously selected triplets. Therefore, using this selection criterion, our strategy encourages selecting responses that are diverse relative to the previously annotated data, thereby promoting broader exploration of the prompt-response domain to obtain more informative human preference feedback. This exploration helps improve the implicit reward function by training it on diverse human feedback across domains. Although Proposition 1 is derived for a fully connected neural network, we argue in Section A that its conclusions extend to the transformer architecture used in our experiments.

The matrix $V_{t-1}^{-1}$ in Eq. (3) plays an important role as a *diversity regularizer*: it down-weights directions in the gradient space that have already been well-explored by previously selected data points. Specifically, as more data points are selected, $V_{t-1}$ accumulates the outer products of their gradient features, causing data points with gradient features similar to those already selected to receive smaller values of the selection criterion, and thus be deprioritized. This encourages selecting data points whose gradient features are diverse and complementary to those previously selected, promoting broader coverage of the gradient space. However, directly computing and storing the matrix $V_{t-1}$ (and its inverse) is intractable for modern LLMs, as the gradient dimension equals the number of model parameters. We address this computational challenge in Section 3.3 through LoRA gradients and random projection, which reduce the gradient dimension to a tractable size (e.g., 8192).

In addition to being theoretically grounded, our selection strategy offers two practical advantages. First, our uncertainty criterion is defined using the LLM being trained, rather than external models used in existing methods (Carvalho Melo et al., 2024; Das et al., 2024). Defining the uncertainty criterion independently of the LLM implicitly assumes that different LLMs require the same data for preference alignment, an assumption that does not hold in practice, as we demonstrate empirically in Section 4. Our selection strategy is therefore model-specific, enabling it to identify and select data that is better tailored to aligning the target LLM with human preferences, leading to more effective preference optimization. Second, our selection strategy directly targets data that improves the implicit reward function defined by the LLM and, consequently, its generation quality, through the DPO objective. This contrasts with prior work that selects data to improve a separate reward model used in RLHF, which subsequently requires an additional reinforcement learning step to obtain the final LLM from the learned reward model. This two-stage dependence introduces a potential misalignment between the data selection objective and the final alignment goal: even if the reward model improves, the subsequent RL step may fail to fully capitalize on this improvement, meaning that selected data points may not translate directly into better alignment performance of the final aligned LLM.

## 3.3 PRACTICAL CONSIDERATIONS

Our selection criterion in Eq. (3) requires the computation of gradients of the implicit reward function with respect to the LLM parameters for each prompt-response pair, as well as updating the LLM using the DPO training in every iteration. These steps are computationally expensive and require significant storage for gradient updates. To address these inefficiencies, we propose two acceleration techniques to improve the efficiency of our selection strategy, which we describe in detail next.

**Batch selection.** In each iteration, we select a batch of $B$ prompt-response triplets for human labeling, with the total number of iterations set to $T = k/B$ to maintain the annotation budget. The batch selection accelerates the selection in two ways: 1) We only need to recalculate the gradient for each prompt-response pair (i.e., $\nabla r_{\theta_{t-1}}(x, y)$) every $B$ selections of data instead of every selection; 2) We only need to update the model via DPO training every $B$ selections.

Batch selection dramatically reduces the computational cost of our selection strategy; however, at the expense of information loss. Specifically, the data selected in the current batch will differ from previous batches, but the selection within the current batch may not yield similar results to those in previous batches. To remedy this, we propose to update $V_{t-1}$ within the batch. Specifically, after a data point is selected, we update $V_{t-1}$ using the new data point $(x_b^t, y_{b,1}^t, y_{b,2}^t)$

$$V_{t-1} = V_{t-1} + \varphi_{t-1}(x_b^t, y_{b,1}^t, y_{b,2}^t)\varphi_{t-1}(x_b^t, y_{b,1}^t, y_{b,2}^t) \,. \tag{4}$$

Consequently, the next data point to be selected will also differ from the current one, even though they are in the same batch, thereby further encouraging exploration.

**LoRA gradient with random projection.** Computing gradients in our selection criterion is expensive and requires substantial storage. Specifically, the full gradient of the LLM is the same size as the LLM's weight matrix, and we need to compute and store gradients for all data points. To reduce both computational cost and storage requirements, we propose using LoRA (Hu et al., 2022) to efficiently compute the gradient. However, the LoRA gradient is $1 - 2\%$ of the full model weight, which still requires substantial storage and computation for our selection criterion. Consequently, we apply random projection to further reduce the gradient to a fixed dimension. This random projection is justified by the Johnson-Lindenstrauss lemma (Dasgupta and Gupta, 2003), which shows that the inner product of the original vector can be approximated by the inner product of the projected vector via random projection. Consequently, we can reduce both computational and storage costs dramatically without sacrificing too much in selection performance (as shown in Section 4). Similar techniques have been used in Xia et al. (2024). The random projection also reduces the computational cost of the matrix inverse in $V_{t-1}$ in our selection criterion.

**Gradient normalization.** Existing work (Xia et al., 2024) has demonstrated that the LLM gradients will have lower magnitudes in their $l_2$ norms when the training data are longer in length (i.e., sentence length). This means if we use the selection criterion defined in Eq. (3), we will have a higher chance of selecting training data with shorter lengths. This is undesirable, especially for question-answering applications, where humans may prefer medium- to long-form answers that provide more elaboration. To remedy this, we propose to normalize all the gradients to the unit norm (i.e., $l_2$ norm being 1)

before we use these gradients to calculate the selection criterion, consequently avoiding the criterion favoring shorter sentences. We have empirically shown the effectiveness of normalization before calculating the selection criterion in Section 4.

### 3.4 COMPUTATIONAL COMPLEXITY OF ACTIVEDPO

We theoretically analyze of the computational complexity and memory requirements of ACTIVEDPO.

Assume that we have $n$ number of prompts with $m$ number of responses, the number of parameters is $k$, and the projection dimension is $d$. Calculating the gradient for all the data points requires $O(nm^2k)$. Projecting all the gradients to $d$ dimensions requires $O(nm^2kd)$, and hence a total of $O(nm^2kd)$. Assume that we have selected $s$ number of data points. The calculation of gradient and projection for these $s$ number of selected data points with the new model is $O(skd)$. Calculating our acquisition function for all data points is $O(nm^2d^2 + d^3)$. Therefore, for each iteration, we have the complexity of $O(nm^2d^2 + d^3 + skd + nm^2kd)$. Note that the projection dimension is a controllable hyperparameter, and $m$ is typically small in most applications. Therefore, the overall computational complexity is small. The memory requirement is mainly dominated by storing the projected gradient, which is $O(nm^2d)$ and is again reducible by reducing $d$ and $m$.

**Asymptotic vs. practical complexity.** We note that the asymptotic complexity analysis above does not directly reflect the practical wall-time of different operations. In practice, the dominant computational cost of ACTIVEDPO comes from the forward and backward passes through the LLM to compute gradients for each data point, rather than the random projection step. Specifically, while the random projection has a higher asymptotic complexity $O(nm^2kd)$, the actual wall-time is dominated by the gradient computation through the LLM, as modern GPU-accelerated matrix multiplications make the projection step relatively fast.

**Computational cost comparison across selection strategies.** We provide a comparison of the computational overhead of different selection strategies in Table 1. Random selection has a negligible selection cost. APO (Das et al., 2024) requires computing features from a separate reward model and solving a linear algebra problem, resulting in moderate overhead. APLP (Muldrew et al., 2024) requires forward passes through the LLM to compute reward estimates for all candidate data points. Our ACTIVEDPO requires both forward and backward passes through the LLM (using LoRA) to compute gradients, followed by random projection and the selection criterion computation. The additional computational cost of ACTIVEDPO is justified by its superior label efficiency, as the cost of human annotation typically far exceeds the computational cost of data selection.

Table 1: Comparison of computational overhead per iteration for different selection strategies. $n$: candidate data points; $k$: LoRA parameters; $d$: projection dimension; $B$: batch size.

| Method | Selection Cost | Extra Storage |
|---|:---:|:---:|
| Random | $O(1)$ | None |
| APO | $O(nk + k^2B)$ | $O(nk)$ |
| APLP | $O(n)$ forward passes | $O(n)$ |
| ACTIVEDPO (Ours) | $O(nkd + nd^2 + d^3)$ | $O(nd)$ |

## 4 EXPERIMENTS

In our experiments, we demonstrate the effectiveness of our selection criterion for selecting data to train an LLM that generates responses better aligned with human preferences. We compare with multiple existing baselines using two widely used LLMs across two preference alignment tasks.

**Datasets.** We consider two tasks that require human preference alignment: 1) TLDR summarization dataset (Liu et al., 2020; Völske et al., 2017), which contains posts from Reddit and the corresponding summarization written by humans; 2) WebGPT dataset (Nakano et al., 2021), which is a long-form question-answering dataset that is marked suitable for human preference alignment. These two datasets contain human preference feedback from annotators and will later be used as an oracle to obtain real human preference feedback.

**Models.** We performance experiments using 3 different LLMs: Llama-2-7B (Touvron et al., 2023), Gemma-2B (Team et al., 2024) and Qwen3-4B (Yang et al., 2025). Using these LLMs shows the

effectiveness of alignment across 3 different model families (i.e., Llama, Gemma, and Qwen) and 3 different model sizes (i.e., models with 7 billion, 2 billion, and 4 billion parameters).

**Baselines.** We compare 4 different selection criteria in our experiments: 1) Random: randomly select data points from the dataset to get human preference feedback; 2) APO (Das et al., 2024): a theoretically grounded method in the setting of RLHF alignment. Their theoretical results are based on the assumption of a linear reward function and are designed for RLHF training; 3) APLP (Muldrew et al., 2024), an active learning method for DPO that uses heuristic uncertainty/certainty quantification to select the data to be labeled; 4) Our proposed method ACTIVEDPO. Note that, for fair comparisons, we only vary how data points are selected for labeling across methods and use the same model training and data labeling pipeline across baselines. Consequently, the only variable that affects performance is the method used to select the data.[1]

**Obtaining human preference feedback.** As new responses are generated by the updated model at each iteration, they are absent from the original preference dataset and therefore lack human preference labels. To make our experiments feasible by avoiding costly human annotations at every iteration, we train a reward model on original human preference feedback and use it as an oracle to automatically annotate newly generated responses with preference labels.[2]

**Evaluation.** The reward model can be used to evaluate how well the LLM aligns its responses with human preferences. To evaluate the performance, we use the trained LLM to generate multiple responses for 100 prompts sampled from the dataset for each task. After that, we use the reward model to compute the average reward across all prompt-response pairs and report the performance. Ideally, if the LLM generates responses that yield higher rewards, it aligns better with human preferences, since the reward model is trained on real human preferences.

**Hyper-parameters.** For each task, we train the initial LLM with supervised fine-tuning with the SFT dataset provided in each task for 1 epoch with the learning rate of $2e - 05$. In each iteration, we randomly select 1000 prompts from the dataset and generate 3 responses for each. Consequently, each prompt will form 3 corresponding triplets $(x, y_1, y_2)$ (i.e., $\binom{3}{2}$ number of pairwise combinations) and hence 3000 data points in the dataset $D_t$. We select 50 data points in each iteration using different selection strategies. We train the model using the DPO objective based on the labeled dataset for 4 epochs with the learning rate of $1e - 4$. As for the LoRA gradient, we use a rank of 128 with $\alpha = 512$. We project all the LoRA gradients to 8192 dimensions, a dimensionality that balances performance and computational costs as we will show later.

**Computational cost vs. label efficiency.** We emphasize that the primary goal of ACTIVEDPO is *label efficiency*: achieving better alignment performance with fewer human annotations. While ACTIVEDPO incurs additional computational costs for gradient computation and data selection compared to simpler methods such as random selection, the cost of obtaining high-quality human preference annotations typically far exceeds computational costs in practice. Therefore, the additional computational overhead of ACTIVEDPO is justified by its superior label efficiency. We provide a detailed computational cost comparison among different selection strategies in Section A.

**Results.** We have provided the comparison of the average reward of the responses generated by the LLM trained on the data selected by different selection strategies in Fig. 1. The LLM trained on data selected by our ACTIVEDPO consistently generates responses with higher rewards than other selection strategies across different LLMs and datasets. Consequently, our ACTIVEDPO outperforms all other baselines in selecting data for a fixed number of labeling budgets[3]. APLP performs well on the Gemma model; however, it performs even worse than random on Llama-2. This is likely due to the heuristic design of the uncertainty quantification method in APLP, which does not work consistently well in different settings. Specifically, APLP uses the difference in estimated rewards for two responses given a prompt as part of its selection criterion. This criterion allows APLP to

---

[1]Note that, for APO, we implement the original algorithm (Das et al., 2024) which does not regenerate responses using the new models.

[2]We use reward models that are publicly available on HuggingFace. Specifically, we use the model from OpenAssistant (2024a) for the TLDR dataset and OpenAssistant (2024b) for the WebGPT dataset.

[3]The result of DPO alignment training is problem-dependent (depending on the dataset and model). For very noisy datasets, it is expected that different active learning methods will perform similarly. For larger models, different selection methods also tend to perform more similarly than for smaller models (which explains Fig. 1 (a) where our method performs similarly to other methods in the last iteration).

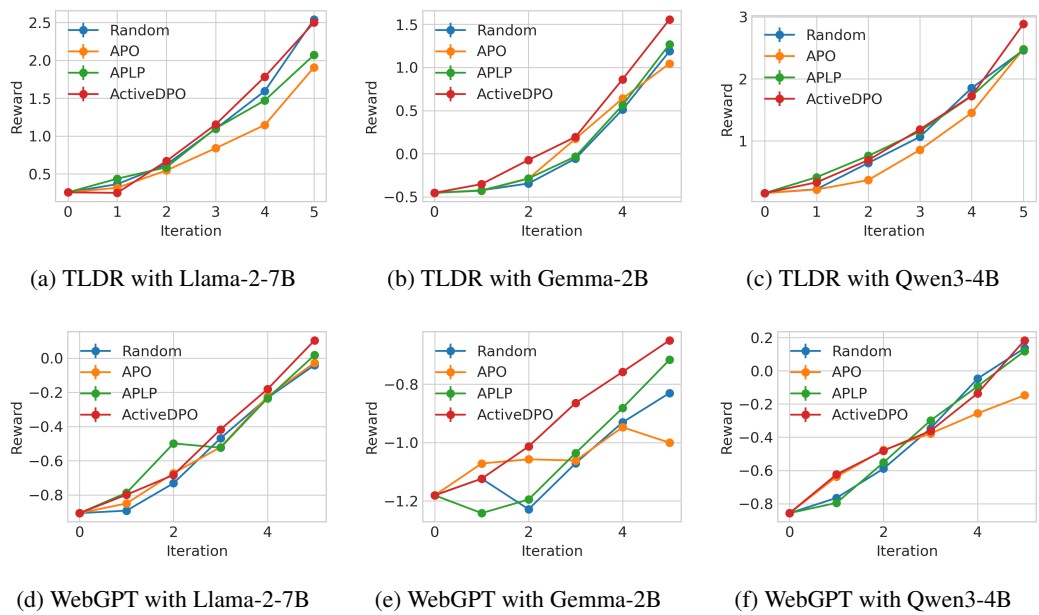

(a) TLDR with Llama-2-7B      (b) TLDR with Gemma-2B      (c) TLDR with Qwen3-4B

(d) WebGPT with Llama-2-7B      (e) WebGPT with Gemma-2B      (f) WebGPT with Qwen3-4B

Figure 1: Average rewards for responses generated by the LLM using different selection strategies.

select triplets with incorrect human preferences predicted by the estimated reward function in the early stage, when the reward function is inaccurate, thereby improving the reward function estimation. This partially explains why APLP performs well in the first iteration for both TLDR and WebGPT on the Llama-2 model. However, as more human preferences are collected, the reward function estimation is more accurate, and hence, the triplet with a large reward difference can be data points with correct human preferences predicted by the estimated reward function and with a large reward margin, which do not help to improve the reward function. Consequently, APLP performs badly in the later iterations. We provide a more detailed discussion on the differences between our gradient-based criterion and APLP's reward-difference criterion in Section A. On the other hand, APO also performs inconsistently in different settings. This is likely due to its unrealistic reward function assumption, which does not hold in practice (e.g., the implicit reward function in DPO is non-linear).

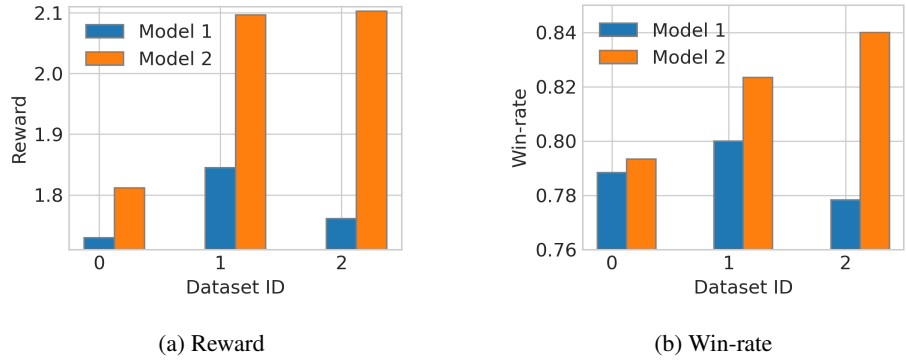

(a) Reward          (b) Win-rate

Figure 2: Different models may require different data to achieve good alignment performance. To verify this, we fine-tune the Gemma model on two SFT datasets to obtain two models, denoted Model 1 and Model 2. We then construct three distinct human preference datasets and perform DPO training on each of these datasets for both models, yielding six fine-tuned models in total.

**The impact of LLM on data selection.** We perform additional experiments to verify that different LLMs require different data to achieve better alignment performance. Specifically, we fine-tune the Gemma-2B model on two different SFT datasets to obtain two distinct LLMs, denoted Model 1 and Model 2. We construct the two SFT datasets by using Sentence-BERT (Reimers and Gurevych, 2019) to embed each data point and then applying $k$-means clustering to partition the full dataset into two

disjoint subsets. We obtain three DPO data subsets using the same procedure. We then train both LLMs on each of the three DPO data subsets and evaluate their alignment performance. As shown in Fig. 2, Model 1 and Model 2 achieve markedly different performance across the three DPO subsets. In particular, Dataset 2 yields the best performance for Model 2 but the worst performance for Model 1 in terms of win-rate, illustrating that the optimal training data is model-dependent. Consequently, the choice of model has a substantial impact on alignment performance and must be taken into account when selecting data for preference optimization. Intuitively, this occurs because Model 1 and Model 2 are trained on disjoint SFT datasets and therefore require different additional data to compensate for the information not covered during their respective SFT stages.

**The effect of random projection on the performance.** Our method uses random projection to reduce the dimensionality of LoRA gradients, thereby reducing memory overhead and computational cost (see Section 3). To further analyze how random projection dimensionality affects the performance of ACTIVEDPO, we evaluate performance across different dimensionalities. The results in Figs. 3c and 3d show that lower dimensionality degrades performance. However, beyond 8192, performance plateaus with no further gain. Therefore, we use a dimensionality of 8192 across all experiments to achieve good performance with lower computational cost and storage requirements.

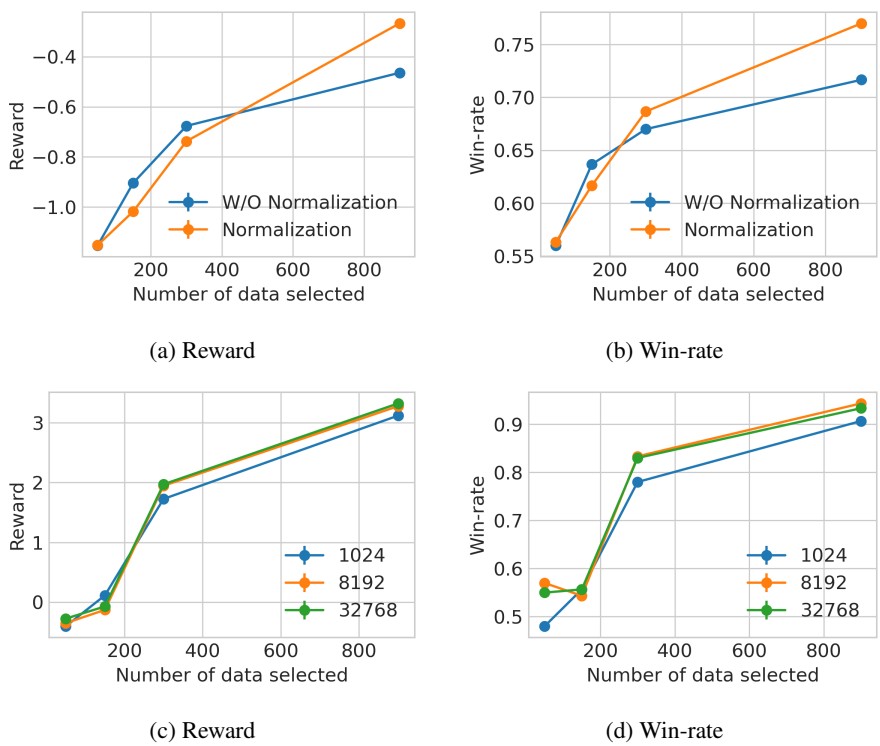

(a) Reward

(b) Win-rate

(c) Reward

(d) Win-rate

Figure 3: **Left two figures:** Effect of normalizing LoRA gradients on the performance of ACTIVEDPO. **Right two figures:** Effect of Random Projection Dimensionality of LoRA gradients.

**The effect of the normalization of the gradient on the performance.** We perform experiments to verify the effect of gradient normalization in ACTIVEDPO. Specifically, as described in Section 3, we normalize the LoRA gradients to unit norm before computing the selection criterion. We compare our gradient-normalized selection strategy against an ablated variant without gradient normalization. Figs. 3a and 3b show the performance of ACTIVEDPO with and without gradient normalization on the WebGPT dataset using the Gemma-2B model. The results demonstrate that normalizing the LoRA gradients consistently improves the performance of our selection strategy. As described in Section 3, without normalization, the selection criterion is implicitly biased toward data points with longer responses, as they tend to produce larger-norm gradients. Since longer responses with clear reasoning are often preferred by human annotators over shorter ones, this bias may appear beneficial; however, it conflates response length with response quality, which is undesirable. Gradient normalization removes this confound, ensuring that data selection is driven by informativeness rather than response

length. We include additional results for the TLDR dataset in Section A, where normalization has a marginal effect on performance, likely due to the more uniform response lengths in that dataset.

## 5  RELATED WORK

Learning from preference feedback has been extensively studied for over a decade (Yue and Joachims, 2009; Fürnkranz et al., 2012; Christiano et al., 2017; Verma et al., 2019; 2020a;b; Zhu et al., 2023; Verma et al., 2025a). In this section, we review work on dueling bandits, active preference learning, LLM alignment, and active LLM alignment, which are most relevant to our problem.

**Dueling Bandits.**  One of the earliest works (Yue and Joachims, 2009; 2011; Yue et al., 2012) considers the finite-armed dueling bandit problem in which the learner's goal is to find the best action using available pairwise preference between two selected actions. Several follow-up works consider different settings involving different criteria for selecting the best action (Zoghi et al., 2014b;a; Ailon et al., 2014; Komiyama et al., 2015; Gajane et al., 2015) and we refer readers to Bengs et al. (2021) for a comprehensive survey covering these details. The standard dueling bandits has been extended to different settings, such as contextual dueling bandit setting (Saha, 2021; Bengs et al., 2022; Di et al., 2023; Li et al., 2024; Verma et al., 2025a).

**Reinforcement Learning with Human Feedback.**  Preference feedback has also been extensively studied in reinforcement learning (Fürnkranz et al., 2012; Akrour, 2014; Christiano et al., 2017; Zhu et al., 2023) introduced preference-based policy iteration, a method that relies solely on preference feedback to guide reinforcement learning, with subsequent developments by (Akrour, 2014). (Christiano et al., 2017) demonstrated the effectiveness of human preference feedback in training agents for Atari games and simulated robot locomotion. On the theoretical side, research has progressed from bandit settings to reinforcement learning (Zhu et al., 2023), providing deeper insights into the optimal use of preference feedback for decision-making and policy optimization. For a more comprehensive overview, we refer readers to a survey on preference-based reinforcement learning (Wirth et al., 2017).

**LLM Alignment.**  Recent works have introduced methods like Reinforcement Learning from Human Feedback (RLHF) (Christiano et al., 2017; Stiennon et al., 2020; Ouyang et al., 2022; Bai et al., 2022; Lee et al., 2024) and Direct Preference Optimization (DPO) (Rafailov et al., 2023) to align LLMs with specific formats or human values. For a comprehensive overview of various aspects of LLM alignment, we refer readers to surveys on the topic (Ji et al., 2023; Anwar et al., 2024).

**Active LLM Alignment.**  Actively selecting preference queries for a human to provide relative preferences between two queries allows for efficient learning of reward functions that capture human intent. Some of the works have already considered actively selecting queries in domains like autonomous  (Sadigh et al., 2017; Biyik and Sadigh, 2018). Recent work on active preference data selection for LLM alignment has explored both heuristic methods (Carvalho Melo et al., 2024; Muldrew et al., 2024) and approaches with theoretical guarantees (Mehta et al., 2023; Das et al., 2024; Verma et al., 2025b). A key distinction among these recently proposed theoretical methods lies in their data selection strategies. The methods with theoretical guarantees either assume a linear latent reward function (Mehta et al., 2023; Das et al., 2024) or do not directly leverage the LLM for active data collection (Verma et al., 2025b), limiting their applicability to the highly non-linear and complex reward functions underlying LLM alignment.

## 6  CONCLUSION

In this paper, we propose a data selection method for actively selecting data to obtain human preference feedback for LLM alignment, aiming to achieve better alignment performance with as few annotations as possible. To this end, we introduce a theoretically grounded method, ACTIVEDPO, and demonstrate that it achieves superior alignment performance under the same labeling budget across different models and datasets. Notably, the selection criterion in ACTIVEDPO requires computing the gradient of the LLM with respect to model parameters for each data point, which is computationally expensive and demands substantial storage for storing gradients. We propose several techniques to improve the efficiency of our method. Although further efforts could accelerate gradient computation, this is beyond the scope of the current work and is left for future research.

## ACKNOWLEDGMENTS

This research is supported by the National Research Foundation, Singapore under its National Large Language Models Funding Initiative (AISG Award No: AISG-NMLP-2024-001). This research is supported by the National Research Foundation (NRF), Prime Minister's Office, Singapore under its Campus for Research Excellence and Technological Enterprise (CREATE) programme. The Mens, Manus, and Machina (M3S) is an interdisciplinary research group (IRG) of the Singapore MIT Alliance for Research and Technology (SMART) centre.

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

# A APPENDIX

## A.1 COMPUTATIONAL RESOURCES, DATASETS AND MODELS

Experiments are run on a server with an AMD EPYC 7763 64-Core Processor, 1008GB RAM, and 8 NVIDIA L40 GPUs.

**Dataset license.** TLDR dataset: MIT License; WebGPT dataset: Apache License 2.0.

**Model license.** Llama-2: LLAMA 2 Community License Agreement. Gemma: Gemma License. Qwen3: Apache License 2.0.

## A.2 ADDITIONAL DISCUSSION ON PROPOSITION 1

Although our results in Proposition 1 rely on neural tangent kernel (NTK) theory, which is primarily developed for fully connected networks, recent works (Lin et al., 2024; Chen et al., 2025) have shown promising directions for partially extending NTK theory to transformers. Furthermore, recent studies (Lin et al., 2024) have demonstrated that sufficiently large transformer models, when pre-trained on offline interaction sequences, can approximate near-optimal online reinforcement learning algorithms such as LinUCB (Li et al., 2010) and Thompson Sampling in multi-armed bandits (Agrawal and Goyal, 2013), as well as UCB value iteration for tabular Markov decision processes (Azar et al., 2017). In addition, transformers have been shown to effectively handle non-stationary RL environments, achieving near-optimal performance by minimizing dynamic regret (Chen et al., 2025).

Our analysis relies on two standard results from Neural Tangent Kernel (NTK) theory: (i) Kernel constancy: along training, the NTK remains (asymptotically) constant (i.e., it converges to a deterministic kernel independent of the training step); (ii) GP limit of the predictor: the trained predictor converges to the Gaussian process induced by that kernel. Result (i) has been established for transformer architectures via the tensor programs framework (Yang, 2020). By contrast, a general proof of (ii) for transformers is not yet available; however, extensive empirical evidence supports Gaussian process behavior in large-width networks (Malladi et al., 2023). Accordingly, the principal theoretical gap in our analysis is a formal proof of (ii) for transformers, which is a challenging problem that we leave as future work. Nonetheless, these assumptions align with existing theory and are corroborated by the strong empirical performance of our method, which together provide a credible justification for applying our theoretical insights to transformer-based LLMs. Equipped with these ideas and existing results, we could potentially extend Proposition 1 to transformer architectures; however, this is beyond the scope of the current paper and is therefore left for future work.

## A.3 COMPARISON WITH EXISTING WORK

**Differences with APLP (Muldrew et al., 2024).** APLP selects data based on the *reward difference* between two responses, i.e., the absolute value of the difference in estimated rewards $|r_\theta(x, y_1) - r_\theta(x, y_2)|$. While intuitive, this criterion only captures the current model's confidence in ranking the two responses and does not account for how labeling a particular data point would improve the model after a parameter update. In contrast, our ACTIVEDPO uses the *gradient difference* $\|\nabla r_{\theta_{t-1}}(x, y_1) - \nabla r_{\theta_{t-1}}(x, y_2)\|_{V_{t-1}^{-1}}$ as the selection criterion, which is inherently more forward-looking: the gradient captures the direction and magnitude of the parameter change that would result from training on a data point. By selecting data points that maximize the gradient-based uncertainty, ACTIVEDPO selects data that would lead to the largest potential improvement in the reward model after a parameter update, rather than simply identifying data points where the current model is uncertain.

Furthermore, APLP's reward-difference criterion can be problematic as training progresses: when the reward model becomes more accurate, data points with large reward differences are likely to be those with correct but highly confident predictions, which provide diminishing returns for model improvement. This partially explains the performance degradation of APLP observed in later iterations of our experiments (Fig. 1). Our gradient-based criterion avoids this issue since it measures informativeness through the lens of the model's parameter space, which remains meaningful regardless of the model's prediction confidence. Additionally, the $V_{t-1}^{-1}$ matrix in our selection criterion serves as a diversity regularizer, accounting for information from previously selected data points. This

ensures that newly selected data points provide complementary information, a property that APLP's selection criterion does not explicitly enforce.

## A.4 ADDITIONAL EXPERIMENTAL RESULTS

We show the win-rate of different selection strategies in Fig. 4. In general, our ACTIVEDPO still outperforms other selection strategies in the last few iterations.

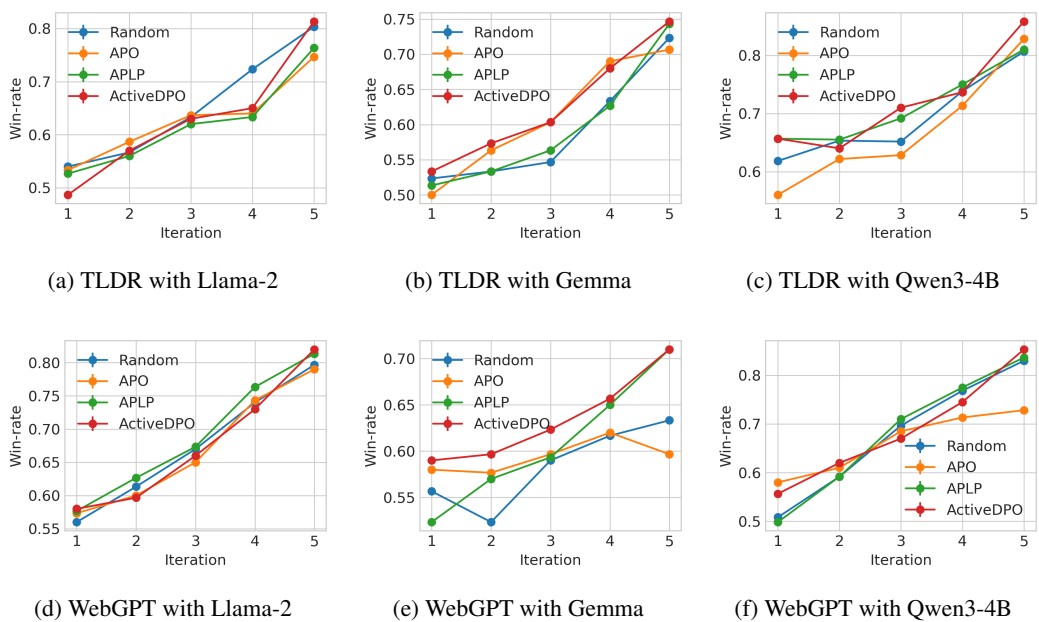

Figure 4: Comparison of the win-rate of the responses generated by the LLM trained by DPO with the responses generated by the initial LLM with different selection strategies.

## A.5 PROOFS FOR PROPOSITION 1

Define the following objective function:

$$L(\theta) = -\frac{1}{m} \sum_{(x,y_w,y_l) \sim D^l} \left[ \log \sigma\big(r_\theta(y_w \mid x) - r_\theta(y_l \mid x)\big) \right] + \frac{\lambda \|\theta - \theta_0\|}{2} . \tag{5}$$

Define $H$ as the NTK matrix following the same definition in Verma et al. (2025a). Define $\nu_T$ following the same definition in Verma et al. (2025a). Define $K$ as the size of the selection dataset $D^s$ in each round.

We make the following assumption:

**Assumption 1.** *Assume that*

- *the reward function $r$ is an unknown, non-linear, and bounded function,*

- $\kappa_\mu \doteq \inf_{x \in \mathcal{X}, y_1, y_2 \in \mathcal{Y}} \sigma(r(x, y_1) - r(x, y_2)) > 0,$

- *the reward function is bounded: $|r(x, y)| \leq 1, \forall x \in \mathcal{X}, y \in \mathcal{Y},$*

- *there exists $\lambda_0 > 0$ s.t. $H \succeq \lambda_0 \mathbf{I}$, and*

- *the reward function takes a vector $z$ (which is the representation vector for the concatenation of $x$ and $y$) as input and $z$ satisfies: $\|z\|_2 = 1$ and $[z]_j = [z]_{j+d/2}$ for all $x \in \mathcal{X}$ and $y \in \mathcal{Y}$.*

Denote $\overline{\sigma}_{t-1}(x, y_1, y_2) = \frac{\lambda}{\kappa_\mu} \|\varphi(x, y_1, y_2)\|_{\overline{V}_{t-1}^{-1}}$ where $\varphi(x, y_1, y_2) = \frac{1}{\sqrt{m}}\big(\nabla r_{\theta_0}(x, y_1) - \nabla r_{\theta_0}(x, y_2)\big)$ and $\overline{V}_{t-1} = \sum_{p=1}^{t-1} \sum_{x, y_1, y_2 \sim D_p^s} \varphi(x, y_1, y_2)\varphi(x, y_1, y_2) + \frac{\lambda}{\kappa_\mu}\mathbf{I}$. We give the following Lemma, which is a direct extension of Theorem 1 of Verma et al. (2025a):

**Lemma 1.** *Given that Assumption 1 holds, let $\delta \in (0, 1)$, $\varepsilon_{m,t} \doteq Cm^{-1/6}\sqrt{\log m}L^3(\frac{t}{\lambda})^{4/3}$ for some absolute constant $C > 0$. As long as $m \geq poly(T, L, K, 1/\kappa_\mu, 1/\lambda_0, 1/\lambda, \log(1/\delta))$, then with probability of at least $1 - \delta$,*

$$\left| \big[r_{\theta_{t-1}}(x, y_1) - r_{\theta_{t-1}}(x, y_2)\big] - \big[r(x, y_1) - r(x, y_2)\big] \right| \leq \nu_T \overline{\sigma}_{t-1}(x, y_1, y_2) + \varepsilon_{m,t}$$

*for all $x \in \mathcal{X}$ and $y_1, y_2 \in \mathcal{Y}, t \in [T]$ when using the objective defined in Eq. (5) to train this reward function $r_{\theta_{t-1}}$.*

*Proof.* This Proposition is immediately true by concatenating the prompt $x$ and response $y$ to replace the input used in Theorem 1 of Verma et al. (2025a) and instantiate the link function in Verma et al. (2025a) as the sigmoid function. Specifically, we assume that the reward takes the representation vector $z$ of the concatenation of $x$ and $y$ as input and assume that this $z$ satisfies the corresponding conditions in Assumption 1. $\square$

Denote $\sigma_{t-1}(x, y_1, y_2) = \frac{\lambda}{\kappa_\mu} \|\varphi_{t-1}(x, y_1, y_2)\|_{V_{t-1}^{-1}}$ where $\varphi_{t-1}(x, y_1, y_2) = \frac{1}{\sqrt{m}}\big(\nabla r_{\theta_{t-1}}(x, y_1) - \nabla r_{\theta_{t-1}}(x, y_2)\big)$ and $V_{t-1} = \sum_{p=1}^{t-1} \sum_{x, y_1, y_2 \sim D_p^s} \varphi_{t-1}(x, y_1, y_2)\varphi_{t-1}(x, y_1, y_2) + \frac{\lambda}{\kappa_\mu}\mathbf{I}$.

**Lemma 2.** *Given that Assumption 1 holds, for some absolute constant $C > 0$, we have that:*

$$|\sigma_{t-1}(x, y_1, y_2) - \overline{\sigma}_{t-1}(x, y_1, y_2)| \leq C\lambda^{-5/6}(t-1)^{4/3}m^{-1/6}\sqrt{\log m}L^{9/2} . \tag{6}$$

*Proof.* Following the proof of Lemma B.4 in Zhang et al. (2021), we can show that

$$|\sigma_{t-1}(x, y_1, y_2) - \overline{\sigma}_{t-1}(x, y_1, y_2)|$$
$$\leq \frac{1}{\sqrt{\lambda}} \left\| \frac{r_{\theta_{t-1}}(x, y_1) - r_{\theta_{t-1}}(x, y_2)}{\sqrt{m}} - \frac{r_{\theta_0}(x, y_1) - r_{\theta_0}(x, y_2)}{\sqrt{m}} \right\|_2$$
$$+ \frac{\tilde{C}^2 L}{\sqrt{\lambda}} \sum_{i=1}^{t-1} \left\| \frac{r_{\theta_{t-1}}(x_i, y_{i,1}) - r_{\theta_{t-1}}(x_i, y_{i,2})}{\sqrt{m}} - \frac{r_{\theta_0}(x_i, y_{i,1}) - r_{\theta_0}(x_i, y_{i,2})}{\sqrt{m}} \right\|_2$$

for some absolute constant $\tilde{C} > 0$. In addition, according to Lemma 3 of Verma et al. (2025a), we have that

$$\|r_{\theta_0}(x, y) - r_{\theta_{t-1}}(x, y)\|_2 \leq C_1 m^{1/3}\sqrt{\log m}\left(\frac{t-1}{\lambda}\right)^{1/3} L^{7/2}, \quad \forall x \in \mathcal{X}, y \in \mathcal{Y}, t \in [T]$$

Consequently, we have that

$$|\sigma_{t-1}(x, y_1, y_2) - \overline{\sigma}_{t-1}(x, y_1, y_2)|$$
$$\leq \tilde{C}^2 \frac{L}{\sqrt{\lambda}}(t-1) \times 2 \times \frac{1}{\sqrt{m}} \times C_1 m^{1/3}\sqrt{\log m}\left(\frac{t-1}{\lambda}\right)^{1/3} L^{7/2}$$
$$= C\lambda^{-5/6}(t-1)^{4/3}m^{-1/6}\sqrt{\log m}L^{9/2}$$

for some absolute constant $C > 0$. $\square$

The result of Lemma 2 says that as long as the width $m$ of the NN is large enough, we can ensure that the difference $|\sigma(x_{t,1}, x_{t,2}) - \overline{\sigma}(x_{t,1}, x_{t,2})|$ is upper-bounded by a small constant. Consequently, we can show the following formal version of Proposition 1.

**Proposition 2** (Formal version of Proposition 1). *Given that Assumption 1 holds, let $\delta \in (0, 1)$, $\varepsilon_{m,t} \doteq Cm^{-1/6}\sqrt{\log m}L^3(\frac{t}{\lambda})^{4/3} + C\lambda^{-5/6}(t-1)^{4/3}m^{-1/6}\sqrt{\log m}L^{9/2}$ for some absolute*

*constant $C > 0$. As long as $m \geq poly(T, L, K, 1/\kappa_\mu, 1/\lambda_0, 1/\lambda, \log(1/\delta))$, then with probability of at least $1 - \delta$,*

$$\left| \left[ r_{\theta_{t-1}}(x, y_1) - r_{\theta_{t-1}}(x, y_2) \right] - \left[ r(x, y_1) - r(x, y_2) \right] \right| \leq \nu_T \sigma_{t-1}(x, y_1, y_2) + \varepsilon_{m,t}$$

*for all $x \in \mathcal{X}$ and $y_1, y_2 \in \mathcal{Y}, t \in [T]$ when using the objective defined in Eq. (5) to train this reward function $r_{\theta_{t-1}}$.*

*Proof.* Combining Lemma 1 and Lemma 2, we get that the proposition is true. $\qquad \square$

**Remark 1.** *Note that the objective function of Eq. (5) is almost the same as Eq. (5), with Eq. (5) scaling the Eq. (5) by a constant and having an additional regularization term. The design of Eq. (5) is for the theoretical results. Empirically, we still use the standard Eq. (2) and adjust the regularization by adjusting the $\beta$ in Eq. (2).*

