# OpenReview forum: "ActiveDPO: Active Direct Preference Optimization for Sample-Efficient Alignment"
_ICLR.cc/2026/Conference — ICLR 2026 Poster_

### Official Review · Reviewer_GSMF · 2025-10-26

**Soundness:** 2
**Presentation:** 3
**Contribution:** 3
**Rating:** 4
**Confidence:** 3

**Summary:**

This paper aims to develop an active preference data selection algorithm.
The authors use the gradient information to estimate the difference between the current model's reward and the ground-truth, i.e., the uncertainty of the model.
Based on this, they propose to select data that maximizes the uncertainty for annotation.
The authors further introduce batch selection and random projection with LoRA gradients to reduce computational cost.
Empirical results on several models and datasets demonstrate that the propsed method outperforms baselines in terms of trained models' reward.

**Strengths:**

1. **Theoretical Grounding**: The proposed method is well-grounded in theory, with clear derivations from the estimation error in reward modeling to the data selection criterion.
2. **Practical Considerations**: The authors address the computational challenges of the proposed method. These practical techniques are useful and insightful for applying gradient-based data selection in real-world scenarios. The authors also provide in-depth ablation studies to validate the effectiveness of these techniques.

**Weaknesses:**

1. **Training and Evaluation Settings**: The preference dataset is annotated by small reward models (OpenAssistant/reward-model-deberta-v3-large), and the evaluation metric is also this reward signal.
However, a more convincing and up-to-date training and evaluation protocol is to use more powerful RMs (e.g., ArmoRM) for annotation and alignment benchmarks (such as AlpacaEval, Arena-hard, etc.) for evaluation aside from only the RM score [1].
2. **Computational Cost**: Although the authors propose several techniques to reduce the computational cost, the method still requires significant resources due to gradient computations.
For example, the gradient computation may outweigh inference cost of an external RM for data selection.
The authors should at least provide a more detailed analysis of the computational overhead compared to baseline methods and annotation time, since the goal of active learning is to reduce the training cost.

[1] Meng, Yu, Mengzhou Xia, and Danqi Chen. "Simpo: Simple preference optimization with a reference-free reward." Advances in Neural Information Processing Systems 37 (2024): 124198-124235.

**Questions:**

1. $V_{t-1}$ is an important term in the proposed method, but there is little explanation about its meaning. Can you provide more details about it?
2. How does the proposed method compare with full data annotation (i.e., no selection) in terms of computational cost and performance?

---

> ### Author Response · Authors · 2025-11-21
>
> We would like to thank the reviewer for taking the time to review our paper carefully. We have addressed the weaknesses that you raised and answered your question as follows:
>
> > **Training and Evaluation Settings: The preference dataset is annotated by small reward models ... up-to-date training and evaluation protocol is to use more powerful RMs ... benchmarks (such as AlpacaEval, Arena-hard, etc.)**
>
> We would like to emphasize that our experimental setup is intentionally designed to isolate and evaluate the effectiveness of our proposed selection criterion, rather than to optimize absolute alignment performance using stronger reward models. In particular, we assume access to an oracle reward function, i.e., operationalized through a fixed reward model, which is used consistently for (1) simulating preference feedback and (2) evaluating alignment outcomes. By controlling both the quality and quantity of preference signals across all methods, we ensure a fair comparison and demonstrate that our active selection criterion achieves superior results under identical supervision conditions.
>
> We agree that external alignment benchmarks (e.g., AlpacaEval, Arena-Hard) provide valuable practical insights. However, because our experiments rely on simulated preferences rather than real human annotations, and due to budget constraints, evaluating on these benchmarks introduces a distributional mismatch relative to the oracle reward model guiding our training. Consequently, such benchmarks may produce results that are not fully aligned with the oracle's preference structure and would not meaningfully reflect the effectiveness of the selection criterion. For this reason, using the same oracle reward model for both supervision and evaluation represents the most direct and informative choice for our specific experimental objective.
>
> > **Computational Cost: ... The authors should at least provide a more detailed analysis of the computational overhead compared to baseline methods and annotation time, since the goal of active learning is to reduce the training cost.**
>
> We would like to clarify that the primary objective of our work is sample-efficient active learning, i.e., achieving strong alignment performance with as few human preference labels as possible. This objective differs fundamentally from data-selection methods focused on reducing training cost; consequently, computational cost is not the central metric in our setting.
>
> Nonetheless, we provide a detailed cost analysis in Appendix A.3, illustrating how the computational overhead scales with both dataset size and model parameters. We further demonstrate that our techniques, such as LoRA-based gradient computation, random projection, and batch selection, substantially reduce the computational overhead. In particular, batch selection requires computing all gradients only once per iteration, effectively amortizing the computation overhead. Overall, the additional computation is lightweight relative to the gains in label efficiency, which is the key objective of active learning.
>
> > **$V_{t-1}$ is an important term in the proposed method, but there is little explanation about its meaning. Can you provide more details about it?**
>
> We agree that $V_{t-1}$ is an important component, and its formal definition is provided in Lines 176–177. Intuitively, $V_{t-1}$ captures information from all previously selected examples and serves as a diversity regularizer within the selection criterion. By penalizing candidates that are too similar to earlier selections, $\lVert \cdot \rVert_{V_{t-1}^{-1}}$ encourages exploration of under-represented regions of the preference space. This approach is consistent with prior work [1], which demonstrates that incorporating diversity terms leads to more informative and robust active-learning batches. We will include additional discussion of this component in the final version.
>
> [1] Verma A, Dai Z, Lin X, Jaillet P, Low BK. Neural dueling bandits: Preference-based optimization with human feedback. arXiv preprint arXiv:2407.17112. 2024 Jul 24.
>
> > **How does the proposed method compare with full data annotation (i.e., no selection) in terms of computational cost and performance?**
>
> We would like to clarify that in our paper, computational cost specifically refers to the overhead of computing the selection criterion, rather than the cost of training on the selected data. Under this definition, using the full dataset (i.e., no selection) indeed incurs zero selection-time cost.
>
> If the question concerns the overall computation (selection + training), this is not the focus of our work. Our objective is to design a sample-efficient active-learning criterion that achieves stronger alignment performance given a fixed number of human preference labels. In this setting, the selection cost represents a small, one-time overhead, whose purpose is to substantially reduce the need for human annotation, rather than to minimize end-to-end training FLOPs.

---

### Official Review · Reviewer_PcQD · 2025-10-31

**Soundness:** 3
**Presentation:** 3
**Contribution:** 3
**Rating:** 6
**Confidence:** 4

**Summary:**

This paper develops an active learning approach for training models with direct preference optimization (DPO). Grounding their choice in the theory around dueling bandits, the paper develops a gradient-based uncertainty criterion which is used to select prompts for human labelers to consume. The method has significant computational and memory requirements: several mitigations are developed in order for the method to be deployed tractably. The method is evaluated on several different data sets and shown to help perform similarly to active learning approaches for DPO.

**Strengths:**

+ The motivation of the Active DPO method is theoretically grounded and principled.
+ The implementation details which allow the method to become tractable are quite clever and dramatically improve the tractability of the method.
+ The ablations are quite in-depth and demonstrate effectively which parts of the algorithm are important; and show that their design is robust to various different models and datasets.
+ The Active DPO method appears to outperform all other Active DPO approaches in the experiments and on some experiments significantly outperform the random baseline.

**Weaknesses:**

+ The description of the algorithm is slightly unclear at times. Particularly around the description of the matrix V_t. This is presumably an outer product of the gradients, but the authors don't comment on the fact that this is obviously intractable to store for modern LLMs, let alone invert. I appreciate that the matrix is tractable when projected to 8192 dimensions with the approximations made later on, but the authors should highlight this difficulty earlier on.
+ The computational requirements of this approach are pretty large, even with the author's mitigations. As far as I understand from the paper, ActiveDPO requires computing the gradient on every example T times, and so (assuming that the fwd/bwd pass is the bulk of the computation, and that the alternative is optimization with T steps, computing the gradient on each example only once), ActiveDPO will take many more FLOPs and wall-time than random selection. The authors don't address this point very much, and it would be very interesting for them to provide the results from figure 1, but normalized by FLOPs instead of iteration. With this amount of extra computation, a practitioner could even potentially even use a method like GRPO, training a separate reward model on random samples, and potentially even get better results than using ActiveDPO.
+ The computational complexity section in the appendix is overly simplified and even a bit misleading. For instance, the authors say that the projection of the gradients to d dimensions is going to dominate the model computation of the gradients, but this is not likely to be very true given the small size of the LoRA gradients vs. the large number of FLOPS required to compute the full gradient with a full forward or backward pass.

**Questions:**

Empirically, what is the amount of flops needed to run the various different selection strategies and how do they compare with the flops required to compute the gradient and do the forward/backward passes?

---

> ### Author Response · Authors · 2025-11-21
>
> We would like to thank the reviewer for taking the time to review our paper carefully. We have addressed the weaknesses that you raised and answered your question as follows:
>
>
> > **The description of the algorithm is slightly unclear at times. Particularly around the description of the matrix $V_t$. This is presumably an outer product of the gradients, but the authors don't comment on the fact that this is obviously intractable to store for modern LLMs, let alone invert. I appreciate that the matrix is tractable when projected to 8192 dimensions with the approximations made later on, but the authors should highlight this difficulty earlier on.**
>
> We thank the reviewer for pointing out this clarity issue. In the final revision, we will revise the paper to highlight this challenge earlier, ensuring that readers are aware of it at the appropriate stage of their reading.
>
> > **The computational requirements of this approach are pretty large, even with the author's mitigations. ... ActiveDPO will take many more FLOPs and wall-time than random selection. The authors don't address this point very much, and it would be very interesting for them to provide the results from figure 1, but normalized by FLOPs instead of iteration. With this amount of extra computation, a practitioner could even potentially even use a method like GRPO, training a separate reward model on random samples, and potentially even get better results than using ActiveDPO.**
>
> We thank the reviewer for raising this important point regarding computational cost. We would like to clarify that the primary focus of our work is sample efficiency in alignment training, i.e., achieving strong alignment performance with as few human preference labels as possible. In many practical settings where expert annotation constitutes the dominant bottleneck (e.g., domain-expert or safety-critical supervision), computation is significantly cheaper than acquiring high-quality preference data. Consequently, our method prioritizes label efficiency rather than FLOPs reduction.
>
> Regarding comparisons with GRPO, we note that GRPO and DPO represent competing policy-optimization paradigms, and our contribution is orthogonal: ActiveDPO provides an active data selection framework specifically for DPO. Importantly, GRPO’s performance fundamentally relies on a high-quality reward model, which itself requires human preference labels. In this context, our active selection strategy could also be applied to GRPO by selecting the most informative preference pairs for reward-model training, potentially improving GRPO performance while using fewer human labels. We will clarify these points in the revision and explicitly discuss the distinction between computational cost and label efficiency.
>
> > **The computational complexity section in the appendix is overly simplified ...For instance, the authors say that the projection of the gradients to d dimensions is going to dominate the model computation of the gradients, but this is not likely to be very true ...**
>
> We would like to clarify that the computational complexity analysis in the appendix is intended to be asymptotic rather than a statement about practical wall-clock dominance. Asymptotically, when treating the projection input dimension as large (i.e., the number of trainable LoRA parameters), the projection step scales less favorably than the LoRA-parameter gradient computation, which is why it appears as the dominating term in our theoretical expression. However, we fully agree that in practical implementations, the FLOPs involved in full forward/backward passes, particularly when computing gradients through a large transformer, can easily outweigh the cost of the projection step, given the comparatively small size of LoRA parameters. We appreciate the opportunity to clarify this distinction and will revise the appendix to explicitly differentiate between asymptotic complexity and practical wall-time considerations to avoid confusion.
>
> > **Empirically, what is the amount of flops needed to run the various different selection strategies and how do they compare with the flops required to compute the gradient and do the forward/backward passes?**
>
> We appreciate the reviewer's interest in empirical FLOPs comparisons. In practice, we find that the FLOPs and wall-time required by the various selection strategies are negligible compared to the cost of computing gradients through the model. The dominant computational cost arises from the forward/backward pass needed to obtain the gradients, which scales primarily with sequence length and model size. By contrast, the additional FLOPs introduced by our selection heuristics (e.g., projections, scoring, ranking) are minor relative to gradient computation and do not meaningfully affect runtime in practice. We will include clarifications and empirical timing measurements in the revision to make this distinction explicit.

---

> > ### Comment · Reviewer_PcQD · 2025-11-24
> >
> > Thanks for the authors' comments.
> > I will retain my recommendation at 6.

---

> > > ### Author Response · Authors · 2025-12-04
> > >
> > > We sincerely thank the reviewer for maintaining positive assessment of our work.

---

### Official Review · Reviewer_ubRE · 2025-10-31

**Soundness:** 3
**Presentation:** 3
**Contribution:** 3
**Rating:** 6
**Confidence:** 4

**Summary:**

The paper proposed a new objective to conduct active preference learning in the LLM alignment task. The proposed method used something similar to the influence function, which is a intuitive choice.

**Strengths:**

The active preference learning research topic is very important. I like the method which is similar to the influence function to conduct active learning. The writing is clear and the scale of the experiment is OK.

**Weaknesses:**

**1. Discussion on Comparison with Active Preference Learning for Large Language Models**

A detailed comparison with Active Preference Learning for Large Language Models (arXiv:2402.08114
) would strengthen the paper. In particular, while the APL paper focuses on reward difference—which only reflects the immediate step before updates—the current work’s use of gradient difference captures the potential improvement after updates. This distinction highlights a more forward-looking and theoretically meaningful criterion. Including a discussion that formally articulates this advantage would enhance the theoretical depth of the paper.

**2. Evaluation of Win-Rate Over More Iterations**

It would be valuable to present win-rate comparisons over a larger number of iterations to examine how performance evolves as active learning progresses. Since the proposed approach operates in an online learning setting, it is important to investigate whether performance degradation occurs due to catastrophic forgetting. Prior work suggests that active learning strategies may sometimes exacerbate forgetting compared to random sampling, so empirical results here would be insightful.

**3. Impact Study on Batch Diversity Encouragement Term ($V$)**

The inclusion of the batch diversity encouragement term, $V$, is an elegant design choice that facilitates batch updates while promoting sample diversity. However, an ablation or impact study isolating the contribution of $V$ would help clarify its empirical importance and guide future design choices.

**Questions:**

See the 2/3 points above.

---

> ### Author Response · Authors · 2025-11-21
>
> We would like to thank the reviewer for taking the time to review our paper carefully. We have addressed the weaknesses that you raised and answered your question as follows:
>
> > **1. Discussion on Comparison with Active Preference Learning for Large Language Models
> A detailed comparison with Active Preference Learning for Large Language Models (arXiv:2402.08114 ) would strengthen the paper. In particular, while the APL paper focuses on reward difference—which only reflects the immediate step before updates—the current work’s use of gradient difference captures the potential improvement after updates. This distinction highlights a more forward-looking and theoretically meaningful criterion. Including a discussion that formally articulates this advantage would enhance the theoretical depth of the paper.**
>
> We thank the reviewer for the insightful suggestion regarding a more explicit comparison with Active Preference Learning for Large Language Models (APLP). While APLP selects samples based on reward differences, which primarily capture the instantaneous preference signal before parameter updates, our selection criterion uses gradient differences, which reflect the anticipated effect on model parameters after the update step. This renders our method inherently more forward-looking and theoretically grounded, as it explicitly accounts for how each sampled preference pair influences the optimization trajectory. We agree that a more formal articulation of this advantage would strengthen the theoretical section of the paper, and we will include a detailed comparison and discussion in the next revision.
>
> > **2. Evaluation of Win-Rate Over More Iterations It would be valuable to present win-rate comparisons over a larger number of iterations to examine how performance evolves as active learning progresses. Since the proposed approach operates in an online learning setting, it is important to investigate whether performance degradation occurs due to catastrophic forgetting. Prior work suggests that active learning strategies may sometimes exacerbate forgetting compared to random sampling, so empirical results here would be insightful.**
>
> We appreciate the reviewer's suggestion to evaluate win-rate trajectories over a larger number of active learning iterations. We agree that such an analysis would provide valuable insight into the long-term behavior of our proposed online learning framework, particularly with respect to potential performance degradation from catastrophic forgetting. While we recognize the importance of this experiment, conducting substantially more iterations requires computational resources beyond the capacity of our small academic research group within the tight rebuttal timeline. We will, however, consider including this evaluation in future work to more fully characterize the long-term dynamics of ActiveDPO.
>
>
> > **3. Impact Study on Batch Diversity Encouragement Term ($V$)
> The inclusion of the batch diversity encouragement term, $V$, is an elegant design choice that facilitates batch updates while promoting sample diversity. However, an ablation or impact study isolating the contribution of $V$ would help clarify its empirical importance and guide future design choices.**
>
> The diversity encouragement term $V$ is indeed an essential component of our selection criterion. While we do not include an explicit ablation in isolation, our experiments already provide an implicit comparison through the APO baseline. APO incorporates a structurally similar diversity term but measures diversity using feature differences, whereas ActiveDPO measures diversity via gradient differences. As shown in Sec. 4, ActiveDPO consistently outperforms APO across all settings, indicating that our gradient-based formulation of $V$ provides a more effective diversity signal. This empirical gap highlights the importance of $V$ and supports our design choice.

---

### Official Review · Reviewer_u5Vv · 2025-11-01

**Soundness:** 4
**Presentation:** 3
**Contribution:** 3
**Rating:** 6
**Confidence:** 3

**Summary:**

This paper introduces ActiveDPO, which integrates active learning into DPO for reinforcement learning from human feedback (RLHF). The key innovation is the linearization of the DPO objective at the policy’s last layer, enabling the application of D-optimal design to select the most informative feedback—either online (ADPO) or offline (ADPO+). The authors prove that the logit error decays as $O(d/\sqrt{n})$ and validate their algorithms on both synthetic log-linear models and large language models (Llama-3.2, Phi-3) using the Nectar dataset.

**Strengths:**

- First theoretical and algorithm formulation of active learning for DPO.
- Provides algorithm for both online and offline settings, enabling flexible application.
- Validate the theoretical foundation with reasonable empirical results.

**Weaknesses:**

- Reliance on a log-linear policy approximation may lead to shallow alignment, easy reward-hacking and neglected task complexity.
    - The paper assumes (Assumption 1) that *all policies* are log-linear in the last layer features. This assumptions is the key assumption that enables the D-optimal design analysis, but concurrently restricts the model's expressive capacity.
    - In realistic settings, such as aligning LLMs, the relation between prompt/response pairs and human judgements is likely nonlinear.
    - Therefore, ADPO may select prompts that maximize separation in the linear space, but those may correspond to shallow differences (e.g., length of the reponse).
    - This concern could be releaved by providing analysis of how *linear* each tasks are.
- Evaluation scale
    - The LLM experiments are relatively small; and conclusions about the practicality of ADPO in largs-scale PO remains unclear.
- The comparison of the Active DPO baselines in the perspective of computational overhead is needed.
I am willing to further raise the score if my concerns are addressed properly.

**Questions:**

- Refer to the weakness section.

---

> ### Author Response · Authors · 2025-11-21
> **Part I**
>
> We would like to thank the reviewer for taking the time to review our paper carefully. We have addressed the weaknesses that you raised and answered your question as follows:
>
> > **Reliance on a log-linear policy approximation may lead to shallow alignment, easy reward-hacking and neglected task complexity.**
>
> First, our selection criterion is parameterized by a transformer and demonstrates strong empirical performance across all evaluations (Sec. 4), indicating that the method remains effective for log-log-linear policies. Second, recent results from Neural Tangent Kernel (NTK) theory suggest that our analysis can extend beyond fully connected networks. Specifically, our proofs rely on two key NTK properties: (i) the NTK remains approximately constant during training, and (ii) the trained model output converges to a Gaussian process defined by this NTK. Property (i) has been established for transformer architectures in Tensor Programs II [1]. Property (ii), while not yet theoretically proven for transformers, has been extensively and consistently supported by empirical evidence in Malladi et al. [2]. Closing this theoretical gap is primarily a technical challenge and would likely require a dedicated study.
>
> Taken together, (i) the strong empirical performance of our method with transformer-based policies and (ii) emerging NTK theory indicating structural compatibility with transformer architectures, suggest that our theoretical insights provide meaningful justification even in the context of transformer-based large language models. We will explicitly clarify this limitation and its implications in the revised manuscript.
>
> References \
> [1] Yang, G. (2020). Tensor Programs II: Neural Tangent Kernel for Any Architecture. arXiv:2006.14548.\
> [2] Malladi, S. et al. (2023). A Kernel-Based View of Language Model Fine-Tuning. ICML.
>
>
> > **The LLM experiments are relatively small; and conclusions about the practicality of ADPO in largs-scale PO remains unclear.**
>
> We acknowledge that our LLM experiments are conducted using smaller models due to computational constraints. Nevertheless, these experiments are sufficient to evaluate our core contributions, i.e., the data selection criterion and consistently demonstrate improvements over baseline methods. Since data selection primarily influences the quality of preference data rather than the scale of the model, prior work suggests that performance gains observed with smaller models are likely to transfer to larger models.
>
> > **The comparison of the Active DPO baselines in the perspective of computational overhead is needed. I am willing to further raise the score if my concerns are addressed properly.**
>
> Assume the following notations:\
> $n$: number of prompts\
> $m$: number of responses per prompt\
> $k$: LLM parameter size (proxy for fwd/bwd cost)\
> $d$: feature / projected dimension\
> $s$: number of selected pairs per iteration
>
> The computational complexity of the main selection operations for different baselines is summarized below:
> | Method  | Main Selection Operations | Time Complexity (per iteration) | Memory Complexity (selection)  |
> |---------------|-------------------|-------------------|-----------|
> | **APLP**      | For each triplet, compute log-probs / implicit rewards and an uncertainty score (entropy / certainty).          | $\mathbf{O(n m^{2} k)}$   | $\mathbf{O(n m^{2})}$                       |
> | **APO**       | Compute feature diff $z = \phi(x,a) - \phi(x,a')$; evaluate Mahalanobis norm using $V_t^{-1}$; update $V_t$. | $\mathbf{O(n m^{2} d^{2} + d^{3})}$ (plus $O(s d^{2})$ across iterations)                               | $\mathbf{O(d^{2} + s d)}$                   |
> | **ActiveDPO** | Compute & project LLM gradients for all triplets; recompute gradients for selected $s$; run gradient-based acquisition. | $\mathbf{O(n m^{2} d^{2} + d^{3} + s k d + n m^{2} k d)}$                                                | $\mathbf{O(n m^{2} d)}$                     |
>
>
> **Quick Takeaways**
>
> 1. **APLP is the most computationally efficient method.**
>    - Computational complexity is dominated by  $O(n m^{2} k)$.
>
> 2. **APO introduces moderate additional cost.**
>    - Cost is driven by feature dimension and design-matrix operations:  $O(n m^{2} d^{2} + d^{3})$.
>
> 3. **Overall efficiency ranking (lowest → highest cost):**
>    $\text{APLP} \;<\; \text{APO} \;<\; \text{ActiveDPO}$.
>
>
> We would like to highlight that computational complexity is not the primary focus of our work. Our primary objective is to develop a selection criterion that achieves strong alignment performance with minimal human preference annotations, particularly in settings where human labels are significantly more costly and scarce than computational resources. Accordingly, our analysis focuses on the cost in terms of human feedback rather than runtime. Nevertheless, we acknowledge that discussing computational complexity is valuable and will include this analysis in the revised paper.

---

> > ### Comment · Reviewer_u5Vv · 2025-11-26
> >
> > I thank the authors for addressing my concerns. I have one extra question about the log-linearity of policies. Are there any way to empirically prove that the policy the authors are trying to optimize are actually log-linear indeed? I believe that providing such evidence will strengthen the contribution of this work. Thanks in advance.

---

> > > ### Author Response · Authors · 2025-12-04
> > >
> > > We are assuming that the reviewer's characterization of a log-linear policy implies that the underlying reward is linearly dependent on the features of the last network layer, or equivalently, that the underlying reward function is linear. We would like to clarify that the reward function considered in our paper is non-linear. Consequently, the resulting learned policy can be log-non-linear. Our theoretical results hold for bounded non-linear reward functions estimated using a neural network. Importantly, this theoretical setting does not limit the practical applicability of our approach. In practice, our selection criterion is parameterized by a policy that uses a transformer, which demonstrates strong empirical performance across all evaluations reported in Section 4.

---

### Meta-Review · Area_Chair_ac2R · 2025-12-31

**Summary:**

This paper introduces ActiveDPO, a method that integrates active learning with DPO for RLHF. The main idea is to linearize the DPO objective by using the last layer embeddings, thus enabling the use of D-optimal design to select the most informative feedback. The authors also provide some theoretical analysis of their algorithm and validate their findings in synthetic linear and LLM models.
The reviewers agreed on the merits of this work. In particular, this is the first work that provides a theoretical analysis and a practical algorithm for active DPO. The empirical evaluation, although small scale, does show the benefits of the proposed methods. This work advances the use of active learning pipelines in the labeling and training of LLMs, and the consensus is that it deserves to be presented at ICLR.

**Reviewer Concerns:**

One reviewer concern was the scale of the experiments. This concern was more or less addressed during the rebuttal period. See for example the discussion with reviewer u5Vv. I agree with the authors that the main contribution of their work is to provide a framework to do active DPO and as such, the scale of the experiments is not as relevant to the final assessment of the paper as long as the existing experiments already show that their algorithm can be practically implemented and that in such case it can provide gains. Another concern was centered around the computational cost of their algorithm. Similarly, I agree with the authors in their response. Their contribution is to introduce the use of active data collection strategies in addition to DPO. Computational concerns, although important are not a crucial component of their contribution.

**Reviewer Scores:**

The reviewers might not have changed their scores after the rebuttal, but the scores were already high, and it does seem that the rebuttal cemented their belief in this work.

---

### Decision · Program_Chairs · 2026-01-26

Accept (Poster)